# ABCB5+ Limbal Epithelial Stem Cells Inhibit Developmental but Promote Inflammatory (Lymph) Angiogenesis While Preventing Corneal Inflammation

**DOI:** 10.3390/cells12131731

**Published:** 2023-06-27

**Authors:** Berbang Meshko, Thomas L. A. Volatier, Karina Hadrian, Shuya Deng, Yanhong Hou, Mark Andreas Kluth, Christoph Ganss, Markus H. Frank, Natasha Y. Frank, Bruce Ksander, Claus Cursiefen, Maria Notara

**Affiliations:** 1Department of Ophthalmology, University of Cologne, 50937 Cologne, Germany; berbang.meshko@uk-koeln.de (B.M.); thomas.volatier@uk-koeln.de (T.L.A.V.); yanhong.hou@uk-koeln.de (Y.H.);; 2TICEBA GmbH, Im Neuenheimer Feld 517, 69120 Heidelberg, Germany; andreas.kluth@ticeba.com (M.A.K.); christoph.ganss@ticeba.com (C.G.); 3RHEACELL GmbH & Co. KG, Im Neuenheimer Feld 517, 69120 Heidelberg, Germany; 4Transplant Research Program, Boston Children’s Hospital, Boston, MA 02115, USA; 5Harvard Skin Disease Research Center, Department of Dermatology, Brigham and Women’s Hospital, Boston, MA 02115, USA; 6Harvard Stem Cell Institute, Harvard University, Cambridge, MA 02138, USA; nfrank@partners.org; 7School of Medical and Health Sciences, Edith Cowan University, Perth, WA 6027, Australia; 8Department of Medicine, VA Boston Healthcare System, Boston, MA 02132, USA; 9Division of Genetics, Brigham and Women’s Hospital, Boston, MA 02115, USA; 10Massachusetts Eye & Ear Infirmary, Schepens Eye Research Institute, Boston, MA 02114, USA; bruce.ksander@schepens.harvard.edu; 11Center for Molecular Medicine Cologne (CMMC), University of Cologne, 50937 Cologne, Germany; 12Institute for Genome Stability in Ageing and Disease, CECAD Research Center, Joseph-Stelzmann-Str. 26, 50931 Cologne, Germany

**Keywords:** limbal epithelial stem cells, cornea, (lymph)angiogenesis, ABCB5

## Abstract

The limbus, the vascularized junction between the cornea and conjunctiva, is thought to function as a barrier against corneal neovascularization. However, the exact mechanisms regulating this remain unknown. In this study, the limbal epithelial stem cell (LESC) marker ABCB5 was used to investigate the role of LESCs in corneal neovascularization. In an ABCB5KO model, a mild but significant increase of limbal lymphatic and blood vascular network complexity was observed in developing mice (4 weeks) but not in adult mice. Conversely, when using a cornea suture model, the WT animals exhibited a mild but significant increase in the number of lymphatic vessel sprouts compared to the ABCB5KO, suggesting a contextual anti-lymphangiogenic effect of ABCB5 on the limbal vasculature during development, but a pro-lymphangiogenic effect under inflammatory challenge in adulthood. In addition, conditioned media from ABCB5-positive cultured human limbal epithelial cells (ABCB5+) stimulated human blood and lymphatic endothelial cell proliferation and migration. Finally, a proteomic analysis demonstrated ABCB5+ cells have a pro(lymph)angiogenic as well as an anti-inflammatory profile. These data suggest a novel dual, context-dependent role of ABCB5+ LESCs, inhibiting developmental but promoting inflammatory (lymph)angiogenesis in adulthood and exerting anti-inflammatory effects. These findings are of high clinical relevance in relation to LESC therapy against blindness.

## 1. Introduction

The cornea is the transparent and avascular “windscreen” of the eye. Loss of corneal transparency is the second most common cause of blindness worldwide. An intact corneal epithelium is necessary for transparency and refraction. This epithelial layer is constantly maintained by a quiescent LESC population located in the basal layer of the limbus, the vascularized junction between the corneal and conjunctival epithelium, a structure regarded as the physical barrier of corneal avascularity. The limbal epithelial cells reside in the basal layer of the limbus, but upon injury or due to normal wear and tear of the corneal epithelium, the cells enter the transient amplifying (TA) state while they migrate to the site where they are needed. ATP-binding cassette, sub-family B, member 5 (ABCB5) recently emerged as a highly specific LESC marker [1,2,3]. Ksander et al. demonstrated that ABCB5 marks LESCs and is required for LESC homeostasis, corneal development, and regeneration. Furthermore, the authors demonstrated that ABCB5 loss of function in ABCB5 knockout mice causes depletion of quiescent LESCs due to enhanced proliferation and apoptosis and results in defective corneal differentiation and wound healing [3]. Based on these data, ABCB5 can be used as a tool to identify the role of LESC populations in neovascularization and inflammation, and the ABCB5KO mouse is a reliable LESC deficiency model.

Corneal cells make a large contribution to the antiangiogenic balance in the cornea. This seems to be due to different mechanisms, but primarily due to receptor decoy mechanisms in the epithelium. Recent reports suggest a proangiogenic genetic profile of limbal epithelial cells when compared to their central corneal counterpart [4], and our laboratory has demonstrated that UV-induced loss of stemness leads to an antiangiogenic phenotype [5,6]. However, the exact mechanisms are not yet fully elucidated, and the role of the limbus and its resident stem cells in them and (lymph)angiogenesis remains elusive [7,8,9,10,11,12,13]. 

Corneal neovascularization is primarily regulated by the VEGF family of proteins [14]. Corneal neovascularization follows the infiltration of leucocytes which secrete VEGF [15,16]. For example, it is known that the corneal epithelium deploys several receptor decoy mechanisms to neutralize potential proangiogenic signals, especially those of VEGF molecules. It is demonstrated that VEGFR-3 acts as a “sink” for its ligands to prevent corneal vascularization [9]. Interestingly, although in the context of LESC deficiency, multiple components including cytokines, chemokines, and growth factors, have been shown to contribute to subsequent neovascularization events [17], the mechanisms employed by the limbal niche and especially how its stem cells contribute to corneal avascularity are largely unknown.

In this study, we aim to address the issue of the role of LESCs in corneal avascularity. We have used the ABCB5KO mouse to model limbal stem cell dysfunction and assess their role in the organization of the normal vascular plexus in resting, developing, and adult corneas, as well as in inflamed and vascularized corneas by using a suture placement model. Moreover, we used ABCB5 expression to enrich ex vivo cultured limbal epithelial cells isolated from human cadaveric donors and assess the paracrine effect of the enriched population on the metabolic activity, scratch wound closure, and tube formation ability of human primary blood and lymphatic endothelial cells. Finally, we have used proteomics analysis to understand the differences between ABCB5 Positive and ABCB5 Negative limbal epithelial cells, especially in terms of their inflammatory and (lymph)angiogenic potential. Overall, a better understanding of the role of the ABCB5-positive LESCs on the corneal (lymph)angiogenic mechanisms will eventually help to elucidate the pathogenesis of blinding neovascular corneal and ocular surface diseases, including LESC deficiency, and to develop new treatment options against them.

## 2. Materials and Methods

### 2.1. Mice

ABCB5KO mice were originally received from the laboratory of NY Frank [3]. These mice were subsequently maintained at the facility of the Department of Ophthalmology, University of Cologne. The ABCB5KO mice were extensively backcrossed to the C57BL/6N substrain, which is the one we use as a WT control, and were bred as homozygous colonies. All animals were treated in accordance with the ARVO Statement for the Use of Animals in Ophthalmic and Vision Research. 

### 2.2. Induction and Quantification of Corneal Hemangiogenesis and Lymphangiogenesis

To induce corneal neovascularization, the established model of suture-induced inflammatory corneal neovascularization was applied as previously described [18]. It was shown that suture placement into the corneal stroma in this model induces early and parallel outgrowth of both blood and lymphatic vessels into the normally avascular cornea, simulating conditions of chronic inflammation [19]. In brief, mice were put under general anesthesia by using a mixture of 120 mg/kg ketamine (Pfizer, Berlin, Germany) and 20 mg/kg xylazine (Bayer AG, Bernburg Germany), and 3 intrastromal 11–0 nylon sutures were placed in the corneal stroma with 2 incursions extending over 120° of corneal circumference each. This results in an inflamed (hem)vascularized and (lymph)vascularized cornea. The outer point of suture placement was chosen near the limbus, and the inner suture point was placed near the center of the cornea equidistant from the limbus to obtain standardized angiogenic responses. Sutures were left in place for 14 days, after which the animals were sacrificed and their corneas harvested for analysis. 

### 2.3. Immunohistochemistry of Neovascularization

For analyses of corneas, whole-mounts were harvested, and mice eyes were enucleated, fixed in PFA 4% for 4 h, rinsed, and incubated overnight in 20% sucrose at 4 °C, then stored at −80 °C. Later, corneas were cut out, and whole-mounts were rinsed in PBS and blocked with 2% bovine serum albumin for 2 h. Staining was performed with rat anti-mouse anti-CD31 (PECAM-1) antibody (1:100; cat. no. 910003, BioLegend, Inc., San Diego, CA, USA), rabbit anti-mouse LYVE-1 (1:100; cat. no. 11-034, AngioBio Co., Vienna, Austria), and anti-CD45 Alexa Fluor 647 antibody (1:100; cat. no. 103123, BioLegend, Inc., San Diego, CA, USA) overnight. The next day, the corneas were washed in PBS and incubated with Alexa Fluor® 488 goat anti-rabbit (1:200; cat. no. A11008, Invitrogen, Waltham, MA USA) and Alexa Fluor® 555 goat anti-rat (1:200; cat.no. A-21434, Invitrogen, Waltham, MA USA) for 1h at RT, then washed and mounted onto slides using fluorescence mounting media (DAKO, Santa Clara, CA, USA). Quantification analysis was performed using digital images taken with an Olympus BX53 microscope (Olympus, Hamburg, Germany) and Cell^f Software (Olympus, Hamburg, Germany). In brief, CD31-positive structures were defined as blood vessels, while LYVE-1-positive structures bigger than 300 Pixels were defined as lymphatic vessels. Vessel coverage in areas was quantified and normalized to the total cornea area to obtain percentage coverage. The vessel number of branching points, endpoints, and sprouts was counted and normalized to the total cornea surface area. Blood vessel sprouts were imaged at 200× with Zeiss LSM 880 Airyscan confocal microscope. Image stacks were analyzed using Imaris software 9.9.1 (Bitplane, Zürich, Switzerland).

### 2.4. Primary Human Limbal Epithelial Cell Harvesting and Maintenance

Primary human limbal epithelial (HLE) cells were cultured in a CNT-57 medium provided by CELLnTEC Advanced Cell Systems (Bern, Switzerland). Sectioned corneo–scleral buttons or limbal rims were treated with a 1.2 U/mL dispase II solution (Sigma) for 2 h at 37 °C. Then, the epithelial cells were gently scraped by using a scalpel and following the limbal border in order to achieve an enriched LESC/progenitor population. The cells were then placed into a T-25 tissue culture flask (Nunc). The cultures were cultured at 37 °C and 5% CO_2_ in air; epithelial colonies emerged 3–5 days following isolation. Once the well was at approximately 80% confluent, fibroblasts were removed by using MACS depletion columns and Anti-Fibroblast MicroBeads human (Order no. 130-050-601, Miltenyi Biotec B.V. & Co. KG, Bergisch Gladbach, Germany).

### 2.5. Maintenance of Human Lymphatic and Blood Endothelial Cells

Primary human dermal microvascular lymphatic endothelial cells (LEC; catalog number C12217, lot number 2010909.1) and blood endothelial cells (BEC, catalog number C12225, lot number 0100505) were purchased from PromoCell (Heidelberg, Germany) and were maintained in a supplemented ECGM MV2 culture medium according to the manufacturer’s instructions. Cells from a single donor were used in each case. The cells were passaged once, reaching 80% confluence by using a Trypsin/EDTA (0.04%/0.03%) solution for 2 min, followed by a Trypsin neutralizing solution (0.05% Trypsin Inhibitor in 0.1% BSA; both by PromoCell, Heidelberg, Germany). The cells were expanded up to passage 8. In each experiment, cells from 3 different passages were used.

### 2.6. ABCB5-Positive and Negative Cell Sorting

Human limbal epithelial cells at passage two were harvested with 0.02% EDTA in PBS. After washing with PBS + Ca/Mg, a batch of cells was cultured as HLE non-sorted cells control. The remaining cells were spun down at 500 g for 5 min and resuspended in HRG (Ringer’s lactate solution containing 2.5% BSA and 0.4% glucose.); then incubated with magnetic beads coated with mouse anti-ABCB5 monoclonal antibody (mAb) (Clone 3C2–1D12) (TICEBA) for 25 min at RT under constant motion. The beads-bound cells (considered HLE ABCB5-positive cells) were magnetically separated from unbound cells (considered HLE ABCB5-negative cells). To detach the ABCB5-positive cells from beads, they were incubated with TrypLE™ Express Enzyme (1×) (Gibco™,Grand islan, NY, USA) for 3 min at 37 °C, 5%CO_2,_ then washed 3 times with HRG solution. HLE ABCB5-positive and negative fractions were cultured until 60 to 90% confluence before using them for the next experiment.

### 2.7. Post-Sorting FACS Analysis

Flow cytometry analyses cells were detached by TrypLE™ Express Enzyme (1×) (Gibco™, Grand islan, NY, USA), counted, and blocked in PBS supplied with 0.5%BSA and 2 mM EDTA for 30 min at 37 °C, 5%CO_2_. After 2 washes the cell suspensions were incubated with 10 μg/mL mouse anti-ABCB5 monoclonal antibody (mAb) (Clone 3C2–1D12) conjugated with FITC or isotype-matched control mAb Mouse IgG1 κ (Isotype Control (P3.6.2.8.1), FITC, eBioscience™) for 1 h at 37 °C, 5%CO_2_; then washed twice with PBS supplemented with 2% FCS and incubated with 7-AAD (Biolegend)1:100 for 5 min at RT before analyzing. Subsequently, the cells were analyzed using a BD FACSCanto™ II (BD Biosciences, Qume Dr San Jose, CA, USA. Forward scatter (FSC) and side scatter (SSC) were used to remove cell debris and doublets, and 7AAD-negative cells were defined as live limbal epithelial cells. The ABCB5+ live population was defined as the fraction of live limbal epithelial cells with the highest contrast against the isotype control. The data were analyzed using the FlowJo (version 10, BD Biosciences, Qume Dr San Jose, CA, USA) software program. 

### 2.8. Collection of Conditioned Media

ABCB5positive fraction and ABCB5negative fractionHLE cells were plated in 6 well plates and in equal seeding densities of 10^5^ cells/well in the corneal epithelial culture media CNT-57 (CellnTech, Bern, Switzerland). The cultures were allowed to grow up to 90% confluence. The cells were washed once with PBS, and the CNT-57 media was replaced with MV2 basal endothelial medium (Promocell, Heidelberg, Germany) supplemented with 2% FBS (basal medium, BM). The produced conditioned media were collected after 24 h, centrifuged at 1500 g to clear the dead cells and debris, aliquoted, and stored at −80 °C for a maximum of 2 months before use for experiments.

### 2.9. Alamar Blue Assay

Cell metabolic activity was evaluated by using the Alamar blue assay (Thermo Scientific, Schwerte, Germany). Specifically, LEC and BEC cultures were plated at a seeding density of 3 K cells/96 well plate well and left overnight in CNT-57 media. Subsequently, the cells were treated with CM from ABCB5 positive fraction or ABCB5 negative fraction HLE cells for 24 h. Subsequently, the cultures were treated for 3 h in 100 μL/well Alamar blue reagent diluted in PBS at a 1/10 ratio. A minimum of 5 technical replicates were used. The plates were read in an Epoch plate reader in absorbance mode at 570 nm and 600 nm (BioTek, Bad Friedrichshall, Germany). The percentage of reduction of the Alamar blue reagent was calculated according to the manufacturer’s instructions. Cell-free wells containing Alamar blue reagent were also measured to be used as a reference/blank. These experiments were repeated with CM from a minimum of 3 different donors and with BECs or LECs from a minimum of two different passages.

### 2.10. Tube Formation Assay

The tube formation assays were performed on Matrigel (Corning, Wiesbaden, Germany) using μ-Slide angiogenesis slides (Ibidi, Planegg/Martinsried, Germany) according to the manufacturer’s instructions. Then, BEC or LEC cells were plated at a cell density of 1 × 104/well in a complete endothelial cell medium. One hour later, the cells fully adhered on the Matrigel, and the full medium was replaced with the conditioned media (*n* = 3). The tube networks formed were photographed after 4 and 16 h using a Zeiss Primo Vert inverted microscope fitted with an AxioCam ERc5s camera (Zeiss, Munich, Germany). The number of branches, loops, and branching points was evaluated using the Image J software (version 1.53, U. S. National Institutes of Health, Bethesda, Maryland, USA). The experiments were performed with LEC and BEC cells of two different passages using supernatants from 3 different donors.

### 2.11. Proteomics Analysis

#### Sample Preparation by SP3

Cell pellets from FACS-sorted cells were lysed in 50 µL SDS in PBS using a SONOPULS mini20 sonifier (Bandelin, Berlin, Germany). Protein content was measured with the Pierce BCS protein assay, and 20 µg was used for protein digestion according to the single-pot solid-phase-enhanced sample preparation [20].

### 2.12. LCMS Data Independent Acquisition

#### Used System

Samples were analyzed on a Q Exactive Exploris 480 (Thermo Scientific, Waltham, MA, USA) mass spectrometer equipped with a FAIMSpro differential ion mobility device that was coupled to an UltiMate 3000 nLC (all Thermo Scientific Waltham, MA, USA). Samples were loaded onto a 5 µm PepMap Trap cartridge precolumn (Thermo Scientific, Waltham, MA, USA) and reverse-flushed onto an in-house packed analytical pulled-tip column (30–75 µm I.D., filled with 2.7 µm Poroshell EC120 C18, Agilent, Santa Clara, CA, USA). Peptides were chromatographically separated at a constant flow rate of 300 nL/min and the following gradient: initial 2% B (0.1% formic acid in 80% acetonitrile), up to 6 and in 1 min, up to 32% B in 72 min, up to 55% B within 7.0 min and up to 95% solvent B within 2.0 min, followed by a 6 min column wash with 95% solvent B. The FAIMS pro was operated at −50 compensation voltage and electrode temperatures of 99.5 °C for the inner and 85 °C for the outer electrode. 

### 2.13. Data Independent Acquisition of Samples

MS1 scans were acquired from 390 *m*/*z* to 1010 *m*/*z* at 15 k resolution. Maximum injection time was set to 22 ms, and the AGC target to 100%. MS2 scans ranged from 300 *m*/*z* to 1500 *m*/*z* and were acquired at 15 k resolution with a maximum injection time of 22 msec and an AGC target of 100%. DIA scans covered the precursor range from 400–1000 *m*/*z* and were acquired in 75 × 8 *m*/*z* staggered windows resulting in 150 nominal 4 *m*/*z* windows after demultiplexing. All scans were stored as centroids.

### 2.14. Data Processing and Analysis

Thermo raw files were demultiplexed and transformed to mzML files using the msconvert module in Proteowizard.

A human canonical Swissprot fasta file (downloaded 26.6.2020) was converted to a Prosit upload file with the convert tool in Encyclopedia 0.9.0 [21] using default settings: Trypsin, up to 1 missed cleavage, range 396 *m*/*z*–1004 *m*/*z*, and charge states 2+ and 3+, default charge state 3 and NCE 33. The CSV file was uploaded to the Prosit webserver and converted to a spectrum library in generic text format [22]. The resulting library (20374 protein isoforms, 28307 protein groups, and 1626266 precursors) was used in DIA-NN 1.7.16 [23] to create a library directly from acquired sample data using the MBR function. The applied settings were: Output will be filtered at 0.01 FDR, N-terminal methionine excision enabled, the maximum number of missed cleavages set to 1, min peptide length set to 7, max peptide length set to 30, min precursor *m*/*z* set to 400, Max precursor *m*/*z* set to 1000, cysteine carbamidomethylation enabled as a fixed modification, double pass search enabled.

### 2.15. Statistical Analysis

Statistical analysis of the experimental data was carried out by using Prism 6.0 software (GraphPad, San Diego, CA, USA). A *t*-test with Mann–Whitney post-test was applied. Results producing a *p*-value lower than 0.05 were defined as statistically significant. We used a minimum of 3 experimental triplicates and repeated experiments at least three times. Cells from at least three cadaveric tissue donors were used to account for biological variability. Error bars displayed in graphs correspond to the standard deviation.

### 2.16. Study Approval and Ethics Statement for the Use of Human Tissue

Human cadaveric corneoscleral rims and buttons, a surplus of transplantation surgery, were only used in case where prior research consent was obtained and in accordance with the Declaration of Helsinki. The tissue had ethics approval (UK Cologne local ethics committee, decision number 19-1602) obtained from the Eyebanks of the Department of Ophthalmology in Cologne, Germany. 

## 3. Results

### 3.1. The Effect of ABCB5 on the Developing and Adult Limbal Vasculature

To assess the effect of limbal stem cell dysfunction via the absence of ABCB5 on the limbal vascular plexus, the immunolocalization of CD31 (marking blood vessels) and LYVE-1 (marking lymphatic vessels) on corneal whole-mounts of mice aged 4 (Figure 1A,B), 8 (Figure 1E,F), and 26 weeks (Figure 1G,H) was evaluated. The flat-mount staining of 4-week-old corneas exhibited fewer sprouting vessels in WT animals compared to the ABCB5KO (Figure 1A,B, sprouts, branching points, endpoints, and blood sprouts highlighted with white arrows, yellow arrows, white arrowheads, and blue arrows, respectively). Conversely, this was not observed in the adult animals (8 weeks or 26 weeks, Figure 1E–H for WT and ABCB5KO, respectively).

The morphometric analysis data showed that there was a statistically significant reduction in the total blood vessel area in mm^2^ of both WT and KO animals at the 8-week and 26-week timepoints compared to the 4-week timepoint (WT: 4 weeks 8.8+/−1.7, 8 weeks 6.4+/−1.5 and 26 weeks 6.3+/−1, KO: 8.3+/−1.9, 8 weeks 6.7+/−1, and 26 weeks 6.4+/−1, Figure 2A). The number of lymphatic vessel sprouts per mm^2^ was significantly higher in both KO and WT animals at 4 weeks compared to their 8-week-old counterparts (4 weeks: WT 1.17+/−0.49; KO 1.52+/−0.43 and 8 weeks: WT 0.74+/−0.26; KO 0.93+/−0.25, Figure 2C). In ABCB5KO animals, the difference between sprouts/mm^2^ in 4 weeks and 26 weeks was also significant (4 weeks: KO 1.52+/−0.43 and 26 weeks: KO 0.97+/−0.2 Figure 2C). The number of lymphatic vessel endpoints per mm^2^ was significantly higher in both KO and WT animals at 4 weeks of age compared to their respective 8 weeks and 26 weeks counterparts. Interestingly, there were mildly but significantly more endpoints per mm^2^ in the KO animals compared to their WT counterparts at the 4-week and the 8-week time points (4 weeks: WT 1.98+/−0.5 and KO 2.47+/−0.58, 8 weeks: WT 1.16+/−0.21 and KO 1.69+/−0.41, 26 weeks: WT 1.36+/−0.27 and KO 1.37+/−0.23, Figure 2D). The number of lymphatic vessel branching points per mm^2^ was significantly higher in both KO and WT animals at 4 weeks of age compared to their 8 weeks and 26 weeks counterparts while, similarly to the endpoints, there were significantly fewer branching points per mm^2^ in the KO animals compared to their WT counterparts at the 4-week time point (4 weeks: WT 0.44+/−0.16 and KO 0.7+/−0.24, 8 weeks: WT 0.33+/−0.11 and KO 0.40+/−0.12, 26 weeks: WT 0.26+/−0.16 and KO 0.24+/−0.07, Figure 2E). Furthermore, the number of blood vessel sprouts per mm^2^ was significantly lower in WT compared to the ABCB5KO in 4-week-old mice, but no difference was observed at 8 weeks and 26-week time points (4 weeks: WT 2.34+/−1.23, and KO 8+/−1.34; 8 weeks: WT 1.25+/−0.88 and KO 1.61+/−1.18, 26 weeks: WT 1.25+/−1.12 and KO 1.25+/−0.88, Figure 2F). Table 1 summarizes the key changes in corneal vascularization in ABCB5KO mice at different time points. To depict the sprouts expanding from the blood vessels in the ABCB5 KO, Zeiss Airyscan confocal microscopy followed up by image analysis producing a three-dimensional (3D) representation of the vessels using Imaris V9.9.1 was used (Figure 1I–L). The representative images illustrate examples of these fine extensions, which were found to be more abundant in the ABCB5KO animals (Figure 1I,L for 2D and 3D, respectively). Notably, the number of fine extensions emerging from the blood sprouts was significantly decreased in WT compared to KO mice at the 4-week time point (4 weeks: WT 0.25+/−0.4, KO 2.7+/−1.42, Figure 2F).

### 3.2. The Influence of ABCB5 in Suture-Induced Corneal Neovascularization

To address the question of whether ABCB5 affects the cornea (lymph)angiogenic response following an inflammatory stimulus, we used an established corneal neovascularization model by suture placement on 8-week-old mice [24].

This model typically features a vigorous outgrowth of both blood and lymphatic vessels from the limbal vascular plexus and is regularly used to generate a “high-risk” setting to study corneal transplantation [18]. New blood vessels reached the sutures at 1 week after surgery, and the peak of neovascularization was reached at 2 weeks, which was the experimental endpoint. Corneal flat-mounts of WT (Figure 3A,C) and ABCB5KO (Figure 3B,D) were stained for CD31 and LYVE-1, to assess the extent and morphometric characteristics of the emerging blood and lymphatic networks, respectively, and the morphometric differences between naïve and inflamed corneas (Figure 3A–D, respectively).

The evaluation of flat-mount immunofluorescence images indicated that there was a significant difference in blood and lymphatic vessel areas in mm^2^ between naïve and inflamed mice in both ABCB5KO and WT mice (blood vessels, naïve: WT 10.2+/−1, KO 10.2+/−1, inflamed: WT 14.5+/−3.3, KO 14.4+/−2, Figure 4A, lymphatic vessels, naïve: WT 2.8+/−0.3, KO 2.7+/−0.7, inflamed WT 3.8+/−0.5, KO 4.3+/−0.4, Figure 4B). The same was observed for lymphatic vessel sprouts per mm^2^ (naïve: WT 0.86+/−0.15, KO 0.71+/−0.14, inflamed: WT 2.22+/−0.21, KO 1.57+/−0.61, Figure 4C), endpoints per mm^2^ (naïve: WT 1.53+/−0.22, KO 1.35+/−0.48, inflamed: WT 2.75+/−0.55, KO 2.75+/−0.55, Figure 4D) and branching points per mm^2^ (naïve: WT 0.29+/−0.77, KO 0.3+/−0.94, inflamed: WT 1.15+/−0,26, KO 0.88+/−0.34, Figure 4E). Interestingly, significantly more lymphatic vessel sprouts per mm^2^ were found in the inflamed WT animals compared to their inflamed KO counterparts (WT 2.22+/−0.21 and KO 1.57+/−0.61).

At the same time point of two weeks following suture placement, the cornea whole-mounts were also stained for CD45, a marker of leucocytes, in order to evaluate how ABCB5 affects immune cell infiltration in naïve and inflamed corneas (Figure 5A–D). The percentage of the cornea area covered by CD45+ cells was quantified by using Cell^F. Notably, the fold change in total area covered by CD45+ cells in inflamed eyes normalized to naïve eyes was significantly higher in ABCB5KO animals in comparison to their WT counterparts (total area, WT 1.77 +/− 0.055, ABCB5KO 1.931 +/− 0.045, Figure 5E).

### 3.3. The Absence of ABCB5 in Human Limbal Epithelial Cells Alters Their Paracrine Activity on Blood and Lymphatic Endothelial Cells

In order to study the effect of ABCB5 on the ability of putative LESCs to affect blood and lymphatic endothelial cells *in vitro*, a magnetic cell sorting method was used to enrich for the ABCB5-positive limbal epithelial cells (ABCB5 pos) and separate them from their negative counterparts (ABCB5 neg). Figure 6A depicts a representative histogram of limbal epithelial cell post-sorting FACS analysis for ABCB5 expression, where the isotype control is shown in black, the non-sorted sample in red, the negative fraction in blue, and the positive fraction in green. An enrichment of ABCB5+ cells at a percentage of 63.65% +/− 16.51 on average was achieved at the positive fraction (ABCB5 pos) (Figure 6B). The negative fraction (ABCB5 neg) contained only an average of 3.16% +/− 1.38 ABCB5+ cells, while the non-sorted fraction was 6.81% +/− 3.42 ABCB5+ cells (Figure 6B).

Subsequently, the sorted cells were cultured in order to produce conditioned media by ABCB5 pos and ABCB5 neg fractions. The conditioned media was used to treat blood and lymphatic endothelial cells (annotated BECs and LECs, respectively) to test the responses to metabolic activity via an Alamar Blue assay, their scratch wound closure rate, and their tube formation ability. The Alamar blue assay data demonstrated a significant stimulatory effect of ABCB5 pos cells on BEC cells (ABCB5 pos: 32+/−0.37, ABCB5-: 27.7+/−0.62, Figure 6D) while the LEC metabolic activity was unchanged (Figure 6C). Similarly, the ABCB5 pos cells significantly accelerated scratch wound healing at 16 h time point for LECs (ABCB5 pos: 95.95+/−3.02 and ABCB5 neg: 57.16+/−30.44, Figure 6E) and at both 4 h and 16 h time points for BECs (4 h ABCB5+: 66.5+/−17.3 and ABCB5 pos: 35.2+/−18.4, 16 h ABCB5 pos: 95.5+/−4.68 and ABCB5 neg: 53.8+/−24.21, Figure 6F). Conversely, the tube formation assay data showed no significant difference between the conditioned media in terms of the number of branches (Figure 7A for LECs, Figure 7D for BECs), branching points (Figure 7B for LECs, Figure 7E for BECs) and loops (Figure 7C for LECs, Figure 7F for BECs).

### 3.4. Proteomics Analysis: Donor Variance and Functional Distribution of Identified Proteins

A proteomics analysis comparing ABCB5 positive fraction (ABCB5 pos) versus ABCB5 negative fraction (ABCB5 neg) cultured human limbal epithelial cells from three cadaveric donors was carried out to assess the properties of the limbal stem cell enriched ABCB5+ with a special focus on their role in angiogenesis and inflammation. The principal component analysis (PCA) was generated in Perseus, and component 1 was plotted against component 2. The samples from 3 donors are annotated as ABCB5 pos1, ABCB5 pos2, and ABCB5pos3 (ABCB5-positive fraction in blue) and ABCB5 neg1, ABCB5 neg2 and ABCB5 neg3 (ABCB5-negative fraction in brown). The three ABCB5 pos samples are clustered close together, indicating a more homogeneous population with lower variance compared to their respective ABCB5 neg counterparts (Figure 8A). Taking into consideration a positive *t*-test difference value as well as the statistical significance of *p* < 0.05 led to the identification of 606 proteins that were significantly different between ABCB5 pos and ABCB5 neg cells. The pie chart in Figure 8B illustrates that 560 proteins were significantly upregulated (red) while 46 proteins were significantly downregulated (green) in the ABCB5 pos group. Grey indicated non-significant differences in 6291 proteins, while 144 proteins were not identified (in black). A two-way hierarchical clustering of differentially expressed proteins was generated in Perseus, using Euclidean distance and standard parameters; the analysis identified clear differences in the expression profiles in ABCB5 neg vs. ABCB5pos samples for donors 1, 2, and 3 (Figure 8C). The pie-charts in Figure 8D indicate the number of proteins grouped in the different pathways according to the Reactome database are either downregulated in the ABCB5 pos (top) or upregulated in the positive fraction (ABCB5 pos) (bottom). The largest group of proteins downregulated in the ABCB5 pos samples was the metabolism of proteins (15.1%), followed by disease (12.1%) and metabolism (10.6%). In terms of upregulated proteins, the largest group affected was the metabolism of proteins (15.2%), followed by the immune system group (10%) and the disease group (9.8%). Sixty-two upregulated proteins were involved in signal transduction (9.5%), 59 in the metabolism of proteins (9.1%), and 40 in the metabolism of RNA (6.15%). The groups of gene expression and transport of small molecules were also affected (6.15% and 6.0%, respectively).

### 3.5. Differentially Expressed Proteins Involved in Inflammation and Angiogenesis

We have used the Reactome database to identify candidate proteins that are differentially expressed in the ABCB5 pos and ABCB5 neg populations. Presented in Figure 9A,B, are the proteins with significant *t*-test differences (*p* < 0.05). A positive student’s *t*-test difference suggests that a protein is upregulated in the ABCB5+ population, whereas a negative one indicates that it is downregulated.

By using this strategy, 29 and 17 proteins linked to inflammatory and angiogenesis processes, respectively, were identified as significantly different between the two groups (Figure 9A,B). The distribution of these targets according to Reactome is shown in the pie charts depicted in Figure 8D, corresponding to inflammatory and angiogenesis-enriched pathways, respectively. In both cases, the majority of the detected proteins are involved in the immune system, metabolism, and metabolism of proteins, signal transduction, and hemostasis. 

In terms of proteins involved in inflammatory processes, the ABCB5 pos fraction features an upregulation in anti-inflammatory and inflammation-attenuating molecules, including interleukin-1 receptor antagonist protein, lysosomal Pro-X carboxypeptidase, integrin beta-6, histamine H1 receptor as well as the anti-microbial mediator Protein S100-A8 (Figure 9A). 

On the other hand, Heme oxygenase 1, the proangiogenic regulator’s Heme oxygenase 1, Prostaglandin G/H synthase 2, and Ephrin type-A receptor 1 were upregulated in the ABCB5+ fraction while Thrombospondin 1, an antiangiogenic molecule which plays a key role in cornea avascularity was downregulated (Figure 9A,B).

## 4. Discussion 

LESC injury or dysfunction leads to changes on the cornea surface which are detrimental to vision and are characterized by opacification, neovascularization, defective wound healing, and inflammation. The absence of LESCs in congenital aniridia results in blindness during early childhood [25]. The exact function of LESCs in maintaining corneal avascularity and preventing inflammation remains elusive, especially as their niche, the limbus, is considered a barrier to blood and lymphatic vessels as well as immune cells, but on the other hand, is in direct contact with a rich network of limbal blood and lymphatic vessels. 

The recent discovery of ABCB5 as a LESC-specific marker has facilitated the study of these cells, especially as it is a surface marker, and magnetic or FACS sorting methods can be used to enrich for a more stem cell-like population [3]. Here, we have used a magnetic cell-sorting approach based on the protocols used during the validation of LESC cultures that have also been used in an international multicenter phase I/IIa clinical trial (NCT03549299) to evaluate the safety and therapeutic efficacy in patients with LSCD [26]. Our aim was to enrich for ABCB5-positive cultured human limbal epithelial cells in order to study their proteomic profile, focusing on (lymph)angiogenesis and inflammation as well as their paracrine effect on blood and lymphatic endothelial cells. In addition, we used an ABCB5KO mouse model [3] to assess the role of ABCB5 in cornea avascularity in development and under injury-induced inflammatory stress leading to cornea neovascularization.

Flat-mount staining of ABCB5KO mice and their WT counterparts demonstrated that the blood and lymphatic vessels, as well as the lymphatic vessel sprouts, endpoints, and branching points, decreased in adult (8 weeks and over) mice compared to 4-week-old animals. Zhang et al. documented lymphatic vessel plasticity in the developing mouse cornea, which exhibits an increasing trend from E18.5 (embryonic day 18.5) and peaking at P10 (pups aged 10 days) coinciding with eyelid opening, after which point the lymphatic vessels regress until reaching adulthood [27]. Our laboratory has also shown a reduction of mouse cornea lymphatic vessel area and sprouts in aged mice [18]. This is probably due to decreasing reproducing capacity due to endothelial cell senility, which is also observed in other systems [28,29]. Interestingly, while the blood and lymphatic vessel area was similar between WT and ABCB5KO animals, the lymphatic vessels end and branching points were increased in the KO at 4 weeks and 8 weeks, respectively, indicating a role of ABCB5 while the cornea is in development and in young adult mice. This effect was not observed in adult mice. Moreover, the blood vessel sprouts were more abundant in ABCB5 mice at 4 weeks old but also had a higher number of outgrowing fine extensions. It is already described that the ABCB5KO mouse features aberrant corneal epithelial development with irregular cell morphology, reduced cell numbers, increased proliferation, and apoptosis, as well as reduced expression of pax6 and cytokeratin 12 [3]. These changes, owing to the lack of ABCB5, appear to intensify the lymphangiogenesis phenomenon in the developing cornea (4 weeks); however, the morphometric characteristics of the lymphatic vessels of KO and WT animals are similar in adulthood. Similar findings have been described in epithelial hyperproliferation in destrin-deficient mice [30]. The role of lymphatics as a dynamic surrogate niche has already been shown in the skin, where the lymphatics morphology and spatial proximity change depending on the cycle stage of the hair follicle [28]. It is possible that the lymphatics adapt to the changes that occur in the cornea during development, including the proliferation rate, which peaks at p21 and then continuously reduces until reaching the adult stage [31]. It is possible that the lack of ABCB5, which results in aberrant corneal epithelium proliferation, morphogenesis, and decreased corneal epithelial thickness [3], disrupts the corneal lymphatics during the development leading to a more complex network of vessels; the exact mechanisms under how this is achieved merits further investigation.

Subsequently, the effect of ABCB5 in corneal lymphangiogenesis and leucocyte infiltration following an inflammatory insult in the form of suture placement was evaluated in adult mice. In this case, the total immune cell infiltrate was significantly higher in ABCB5KO animals, and the number of lymphatic vessel sprouts was significantly lower in the ABCB5KO animals suggesting that the presence of ABCB5 in the WT animals promoted lymphatic network complexity. In order to focus on the specific effect of the ABCB5+ limbal epithelial cell population, the effect of magnetic-sorted ABCB5-positive cultured human limbal epithelial cells was compared to their negative counterparts in terms of their paracrine effect on blood and lymphatic endothelial cells. These data indicated a stimulating effect of the ABCB5-positive cells on blood endothelial cell proliferation as well as both lymphatic and blood endothelial cell migration. No significant effect on the tube formation ability and network complexity was demonstrated by the tube formation assays. These results show a previously unreported mild proangiogenic activity of ABCB5-postive limbal stem cells compared to their negative counterparts. Other stem cell types have been previously known to stimulate angiogenesis, including epidermal stem cells [32], mesenchymal stem cells [33,34,35], oral mucosa stem cells [36], and dental pulp stem cells [37,38,39,40]. This seems counterintuitive as the cornea is completely avascular, and the limbus is considered the vessel barrier safeguarding the cornea’s vascular and immune privilege. However, the limbus itself is indeed heavily vascularized and features a delicate network of arcade-forming blood and lymphatic vessels which are thought to provide nutrients and facilitate the passage of growth factors essential for stem cell maintenance [41]. Contrarily, the differentiated epithelium of the peripheral and central cornea expresses antiangiogenic factors that tip the balance towards avascularity, including soluble and membrane-bound VEGFR-3 [9,42,43]. Thus, our data suggest a new concept of a more delicate, context-dependent, and pro(lymph)angiogenic, but anti-inflammatory phenotype of ABCB5-positive limbal stem cells, whereas more differentiated epithelial cells are anti(lymph)angiogenic.

To assess the factors contributing to the proangiogenic effect of ABCB5-positive cells, the proteomic profiles of the magnetic-sorted ABCB5-positive fraction (ABCB5 pos) and the ABCB5-negative fraction (ABCB5 neg) were compared while we focused on differences related to inflammation and angiogenesis factors by using the Reactome software. In terms of angiogenesis, the Reactome revealed several factors upregulated in the ABCB5 pos group that were reported to be relevant to angiogenesis. For example, pro-cathepsin H, in its secreted form, enhances angiogenesis in pancreatic cancer [44], while secreted C-X-C motif chemokine 16 is a positive regulator of angiogenesis by acting as a T-cell attractor [45]. Transforming growth factor-beta-induced protein ig-h3 (TGFBI), a major TGFβ-induced protein relevant to cell adhesion, also present in the serum of patients, promotes colorectal cancer metastasis and angiogenesis [46]. Heme-oxygenase 1(HO-1), an oxidative stress-induced enzyme previously found in stressed corneal epithelial cells [47], has a dual role as an anti-inflammatory factor as well as a proangiogenic mediator upstream of VEGF linked to tissue revascularization following injury. In addition, by-products of HO-1, including CO, iron, biliverdin, and bilirubin, exert numerous effects that can also influence angiogenesis in both positive and negative ways [48]. Therefore, the anti-inflammatory effects of HO-1 can attenuate the excess formation of blood vessels in inflammatory angiogenesis. ABCB5+ cells also feature upregulated prostaglandin G/H synthase 2 (PGHS-2) or cyclooxygenase 2 (COX2), the expression of which is triggered upon stress in the corneal epithelium and promotes corneal angiogenesis [49,50,51]. In fact, COX-2 inhibitors have shown promise as antiangiogenic therapy in the cornea [52,53]. Ephrin type-A receptor 1 (EPHA1), a receptor tyrosine kinase, which was expressed in the corneal epithelium of neovascularized corneas [54], was also found upregulated in ABCB5+ cells.

Contrarily, Thrombospondin-1 (TSP-1), a key regulator of corneal avascularity, also acts as a key activator of latent TGF-β and serves to promote the immunomodulatory and wound healing functions of TGF-β [55] in the corneal epithelium [56] and is downregulated in ABCB5-positive cells. It is considered an essential corneal antiangiogenic factor owing to its apoptotic effect on endothelial cells [57] as well as by inhibiting inflammatory lymphangiogenesis by CD36 ligation on monocytes [58]. Previous reports have already demonstrated a significantly higher expression of TSP-1 in the corneal epithelium and its BM, especially in the basal layers, compared to the conjunctival epithelium as well as the basal limbus, the location of the ABCB5-positive limbal stem cell population [59,60]. The differential expression of these factors in ABCB5-positive cells composes a proangiogenic profile, which is in concurrence with our observations *in vitro* and *in vivo*. 

In terms of the inflammatory factors, the proteomic evaluation revealed several targets, which were upregulated in the ABCB5-positive fraction. Two pro-inflammatory factors have been identified: Firstly, Gamma-interferon-inducible protein 16 (IFI16) plays a role in the sensing of intracellular DNA in virally infected cells and has also been linked to the death of HIV-infected helper CD4 T cells by pyroptosis, a highly inflammatory form of programmed cell death [61]. IFI16 is expressed in the epithelia of the trachea, gastrointestinal tract, skin, and testis, as well as in lymphoid tissues [61], while its potent pro-inflammatory action, when expressed in keratinocytes, has been reported [62]. Secondly, functional epoxide hydrolase 2 (EPHX2), which promotes macrophage infiltration and maturation [63], and its inhibition is considered a pharmaceutical strategy against inflammatory bowel disease [64]. 

While the pro-inflammatory IFI16 and EPHX2 are increased in the ABCB5-positive fraction, these cells also exhibit upregulation of multiple proteins with a known anti-inflammatory effect. Interleukin-1 receptor antagonist (IL-1RA), for example, has an anti-inflammatory action in the cornea when added exogenously as it blocks IL-1alpha, which mediates bone marrow-derived immune cell infiltration [65]. It has been shown to regulate inflammation in order to maintain homeostasis in other systems, including the retina and in endometrial, intestinal, gastrointestinal, and respiratory epithelial cells [66,67,68,69,70,71]. Integrin beta-6 (ITGB6), which is also more abundant in skin epithelial cells, on the other hand, is essential for keeping the lung and skin void of inflammatory cell infiltrates as ITGB6-/-mice develop inflammatory cell infiltrates in these tissues [72]. Antithrombin III, a small glycoprotein that inactivates several enzymes of the coagulation system, has been proposed as an anti-inflammatory treatment of E-coli infection and liver injury [73,74,75] as well as uveitis in a rat model [76]. Similarly, Lysosomal acid lipase/cholesteryl ester hydrolase (LAL) expression in lung epithelial cells of a Lipa-/-(LAL-deficient) model featuring an increase of tumor growth and metastasis associated with the expansion of myeloid-derived suppressor cells, improved inflammation and metastasis [77]. The ABCB5-positive fraction also has an upregulation of Histamine Receptor 1 (HRH1), which regulates allergic responses in nasal [78] and corneal [79,80] epithelial cells, while non-competitive histamine antagonist (H1-receptor) is proposed for the treatment of allergy affecting the ocular surface [81]. Finally, Protein S100-A8 (S100A8), also upregulated in the ABCB5-positive fraction, is an antimicrobial peptide produced by airway epithelium and functions as a potent and direct regulator of macrophage phenotype and function [82]. These results suggest a potential anti-inflammatory activity of ABCB5-positive limbal epithelial cells. Accordingly, *in vivo*, there was an increased CD45-positive leucocyte recruitment in inflamed ABCB5KO mice compared to WT, indicating that the absence of ABCB5 disrupts the anti-inflammatory effect of the LESC population.

Taken together, the *in vivo* observations on the limbal vascular organization of the ABCB5KO and WT mice during development, adulthood, and under inflammatory stress, the *in vitro* data on the paracrine action of ABCB5+ cells on blood and lymphatic endothelial cells, as well as the proteomic analysis reflects a novel, contextual and dynamic role of ABCB5+ limbal stem cells in their niche, in corneal inflammation and avascularity. Specifically, while the ABCB5+ stem cells exert a mild pro(lymph)angiogenic effect, which is reflected in the fact that the adult WT mice feature less complex lymphatic vessels in response to inflammatory stimulation compared to their ABCB5KO counterparts and the human ABCB5+ cells stimulate blood and lymphatic cell proliferation and migration, their proteomic profile also features an upregulation of proangiogenic modulators. This is not surprising as the limbus itself is vascularized, and the resident stem cells use their proangiogenic effect to sustain the elaborate vessel network that supplies them with nutrients. Notably, we have previously shown that UV-induced limbal epithelial cell differentiation, which features loss of ABCB5 [83], also coincides with a reduction of proangiogenic modulators and a less stimulatory paracrine effect on blood and lymphatic endothelial cells [5,6] also showing that ABCB5-positive putative stem cells have a more-proangiogenic effect compared to their differentiated counterparts. 

To counterbalance their proangiogenic activity, the ABCB5-positive cells feature an upregulation of anti-inflammatory proteins, which may enable them to limit immune cells within the limbal barrier. This also reflects on the immune cell infiltration data under inflammatory conditions where the ABCB5KO animals feature higher coverage of CD45-positive cells. The concept of stem cell populations having both a proangiogenic and anti-inflammatory profile is not novel, as this is considered a key feature of mesenchymal stem cells from the pancreas [84] as well as adipose tissue [85]. This dual role is clinically relevant as cultured ABCB5-positive limbal stem cells are used for the treatment of limbal stem cell deficiency in phase II clinical trials (ClinicalTrials.gov Identifier: NCT03549299) [26]. It is possible that while the ABCB5-positive cells have a superior proliferative and regenerating potential, their anti-inflammatory effect prevents the recruitment of inflammatory cells, including macrophages, which produce potent proangiogenic agents such as VEGFs and thus offset any mild proangiogenic effects, thus leading to successful regeneration of the corneal epithelium with no undesired inflammation or neovascularization. This novel concept of the dual role of ABCB5-positive limbal stem cells within their niche sheds new light on the function of the limbal barrier and has clinical relevance in the context of limbal stem cell transplantation. This is especially relevant in light of the ongoing phase I/II clinical trials using ABCB5-positive cells in patients with limbal stem cell deficiency [26].

## Figures and Tables

**Figure 1 cells-12-01731-f001:**
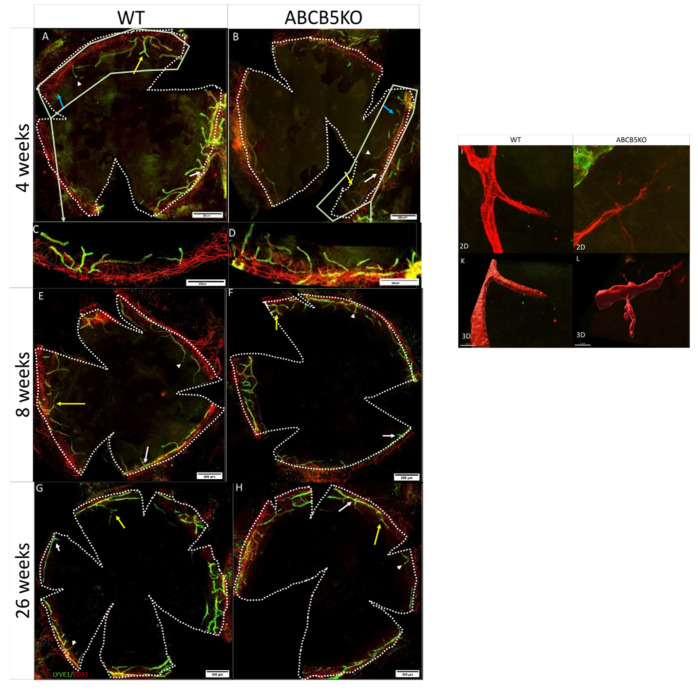
Immunolocalization of CD31 (red, blood vessels) and LYVE-1 (green, lymphatic vessels) on corneal whole-mounts of WT and ABCB5KO mice aged 4 (**A**,**B**), 8 (**E**,**F**), and 26 weeks (**G**,**H**). (**I**–**L**) 400× magnification for blood vessels sprouting in WT and ABCB5 KO 4-week-old mice with thin extensions grow out in the ABCB5 KO, 2D (**I**,**J**) 3D (**K**,**L**). Sprouts, blood sprouts, branching points, and endpoints are highlighted with white arrows, blue arrows, yellow arrows, and white arrowheads, respectively. Representative image showing a closer look at blood vessel morphology (**C**,**D**).

**Figure 2 cells-12-01731-f002:**
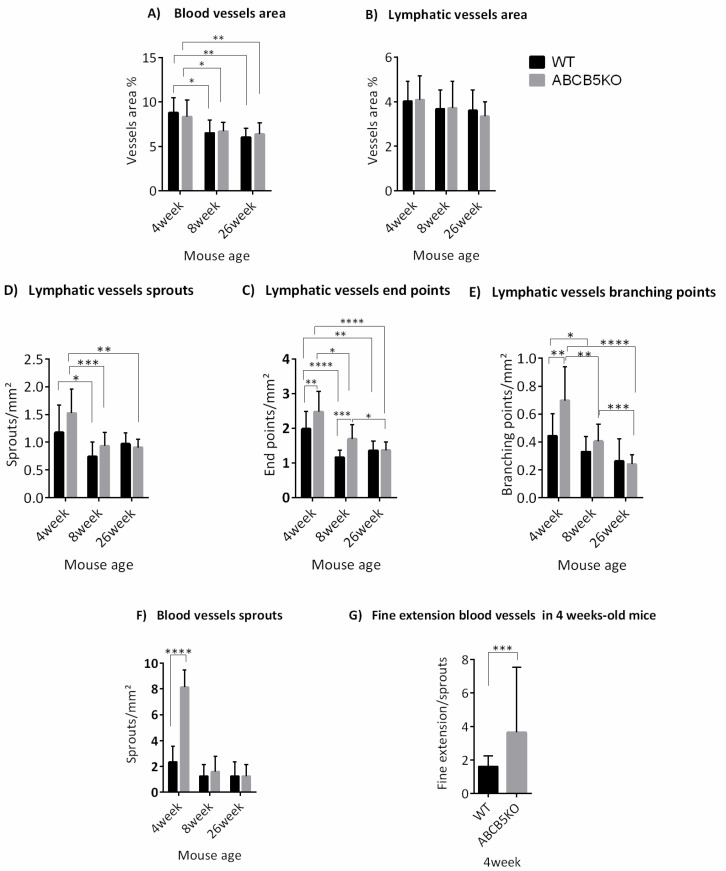
Morphometric analysis data showing means +/− SD of (**A**) the total blood vessel area and (**B**) total lymphatic vessel area in mm^2^ of WT and ABCB5KO at 4-, 8- and 26-week timepoints. The number of lymphatic vessel sprouts per mm^2^ was significantly higher in both KO and WT animals at 4 weeks compared to their 8-week-old counterparts. (**C**) depicts lymphatic vessel sprouts/mm^2^, (**D**) endpoints/mm^2^ and (**E**) branching points per mm^2^. 4 weeks: WT *n* = 17, ABCB5KO *n* = 15, 8 weeks: WT *n* = 19, ABCB5KO *n* = 21, 26 weeks: WT *n* = 9, ABCB5KO *n* = 13. (**F**) Number of blood vessel sprouts per mm^2^ was significantly higher in KO compared to WT animals at 4 weeks: WT *n* = 8, ABCB5KO *n* = 10, 8 weeks: WT *n* = 5, ABCB5KO *n* = 7, 26 weeks: WT *n* = 6, ABCB5KO *n* = 5. (**G**) Number of the fine extension per sprouts was significantly higher in KO compared to WT animals at 4 weeks: WT *n* = 8, ABCB5KO *n* = 10. Asterisks correspond to statistical significance: * *p* < 0.05, ** *p* < 0.01, *** *p* < 0.001, **** *p* < 0.0001.

**Figure 3 cells-12-01731-f003:**
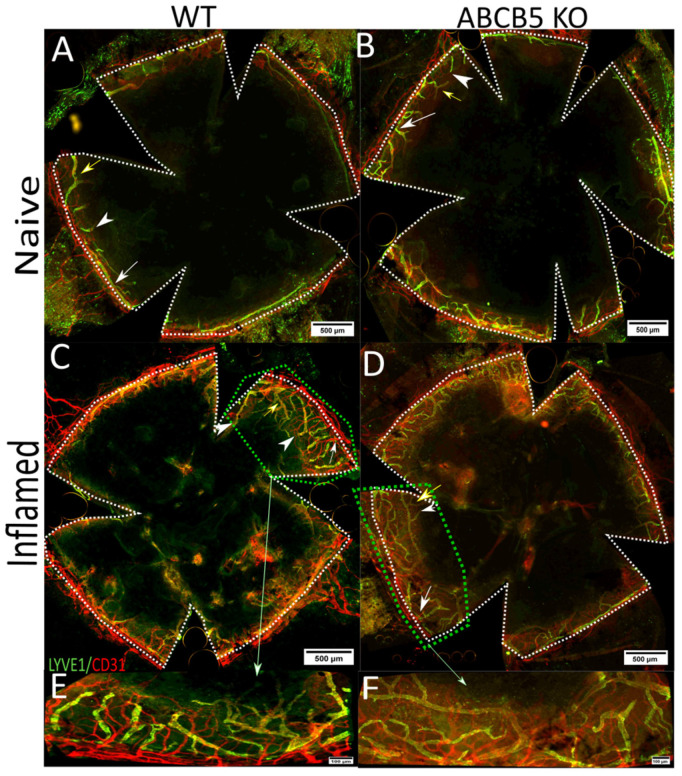
A suture model to induce inflammation and neovascularization; the peak of neovascularization was reached at 2 weeks. Corneal flat-mounts of WT (**A**,**C**) and ABCB5KO (**B**,**D**) were stained for CD31 and LYVE-1 to depict the extent and morphometric characteristics of the emerging blood and lymphatic networks, respectively. Sprouts, branching points, and endpoints are highlighted with white arrows, yellow arrows, and white arrowheads, respectively. Light green arrow and box shows the area with higher magnification (100×). Representative image showing a closer look at blood vessel morphology (**E,F**).

**Figure 4 cells-12-01731-f004:**
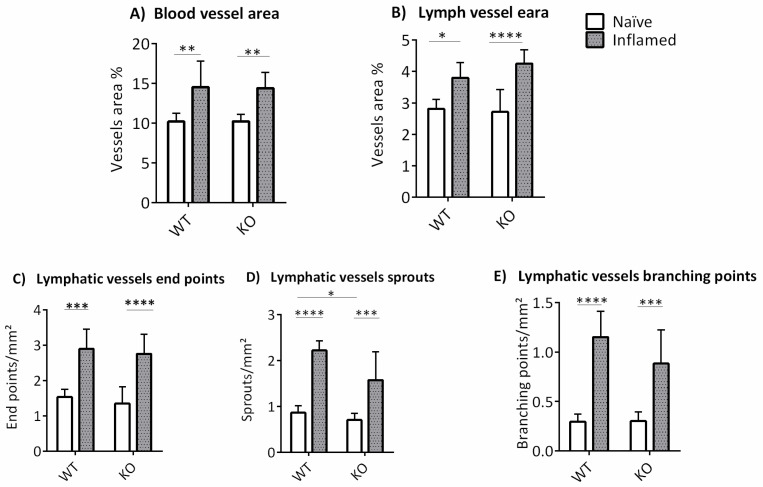
Morphometric analysis data showing means +/− SD of (**A**) the total blood vessel area and (**B**) total lymphatic vessel area in mm^2^ of WT and ABCB5KO 2 weeks following suture placement. Panel (**C**) depicts lymphatic vessel sprouts/mm^2^, (**D**) endpoints/mm^2^, and (**E**) branching points per mm^2^. WT *n* = 6, ABCB5KO *n* = 8. Asterisks correspond to statistical significance: * *p* < 0.05, ** *p* < 0.01, *** *p* < 0.001, **** *p* < 0.0001.

**Figure 5 cells-12-01731-f005:**
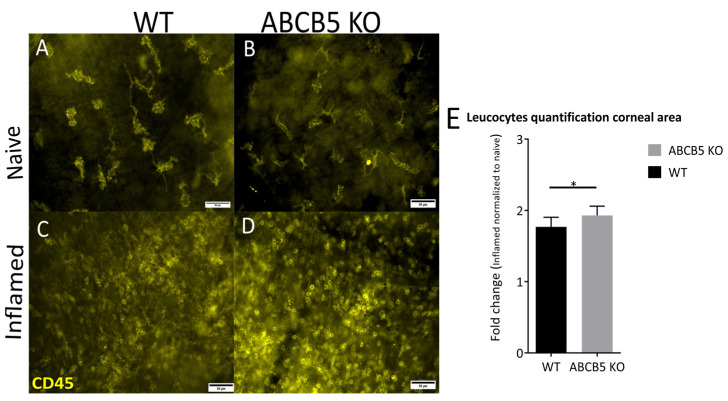
Representative images of CD45+ cells in naïve and inflamed corneal flat-mounts of WT (**A**,**C**) and ABCB5KO (**B**,**D**) (magnification: ×400; scale bar: 50 μm). (**E**) The fold change in total cornealareas covered by CD45+ cells in inflamed eyes was normalized to naïve eyes. WT *n* = 5, ABCB5KO *n* = 5. Asterisks correspond to statistical significance: * *p* < 0.05.

**Figure 6 cells-12-01731-f006:**
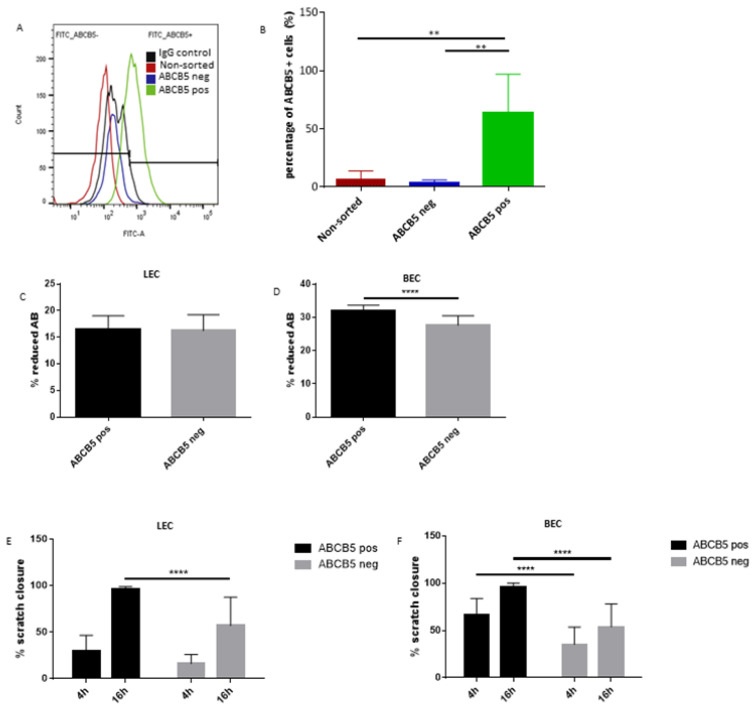
(**A**) Representative post-sorting analysis of magnetically separated cultured human limbal epithelial cells into ABCB5-positive (ABCB5 pos) and ABCB5-negative (ABCB5 neg) cell fractions. The isotype control is shown in black, the non-sorted sample in red, the negative fraction in blue, and the positive fraction in green. (**B**) An enrichment of ABCB5+ cells at a percentage of 63.65% +/−16.51 on average was achieved at the positive fraction (ABCB5 pos). Paracrine effect of ABCB5 pos and ABCB5 neg cells on human blood (BEC) and lymphatic endothelial cells (LEC): Alamar blue assay evaluating the metabolic activity of LEC (**C**) and BEC (**D**) cells 24 h post-treatment and scratch wound assay at 4 h and 16 h time points of LEC (**E**) and BEC (**F**) cells. Graphs feature means +/−SD. Donor number *n* = 3, technical replicates per donor = 3. Asterisks correspond to statistical significance: ** *p* < 0.01, **** *p* < 0.0001.

**Figure 7 cells-12-01731-f007:**
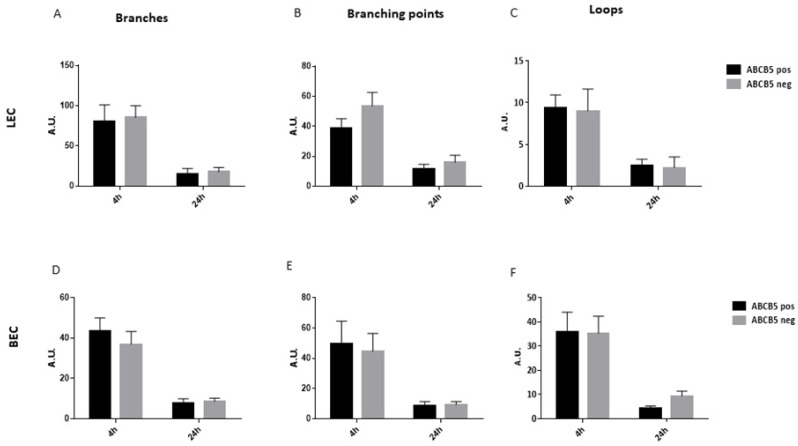
Tube formation assay evaluating the paracrine effect of ABCB5 pos and ABCB5 neg cells on human blood (BECs) and lymphatic endothelial cells (LECs). Evaluation of LECs and BECs branches (**A**,**D**), branching points (**B**,**E**), and loops (**C**,**F**) at 4 h and 24 h time points. Graphs feature means +/−SD. Donor number *n* = 3, technical replicates per donor = 3. No statistical differences were found between any of the groups.

**Figure 8 cells-12-01731-f008:**
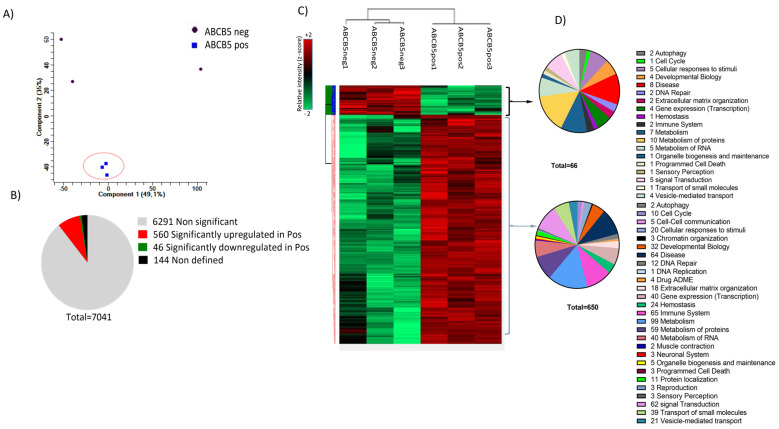
(**A**) Representation of all analyzed samples in two components obtained by principal component analysis (PCA) where component 1 was plotted against component 2 as generated in Perseus; The samples from 3 donors are annotated as ABCB5 pos1, ABCB5 pos2, and ABCB5 pos3 (ABCB5-positive fraction in blue and circled) and ABCB5 neg1, ABCB5 neg2 and ABCB5 neg3 (ABCB5-negative fraction in dark red) The 3 ABCB5 positive fraction samples are clustered close together indicating a population with lower variance compared to the three respective ABCB5-negative fractions. (**B**) A pie chart depicting the number of proteins featuring statistically significant differences (Student’s *t*-test, *p* < 0.05). Grey indicates non-significant differences, while the significantly upregulated and downregulated proteins in the ABCB5positive fractions are shown in red and green, respectively. One hundred forty-four proteins were not defined (in black). (**C**) Hierarchical clustering of differentially expressed proteins, using Euclidian distance and standard parameters as generated in Perseus; average linkage indicates altered expression profiles in ABCB5 negative vs. ABCB5-positive samples for donors 1, 2, and 3. Each row represents a single protein, whereas each column represents a sample. The color scale illustrates the relative individual protein levels across each sample. Green corresponds to downregulated whereas red to upregulated proteins. (**D**) The pie charts show the number of proteins grouped in the different pathways according to the Reactome database are either downregulated in the ABCB5 pos (top) or upregulated in the positive fraction (ABCB5 pos) (bottom).

**Figure 9 cells-12-01731-f009:**
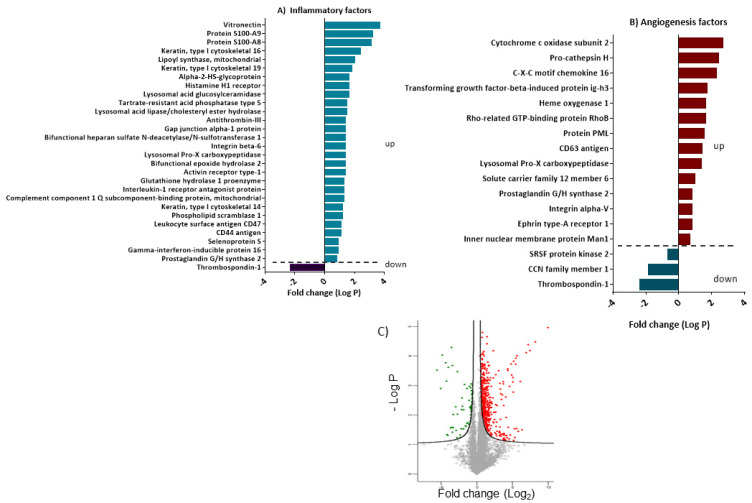
Graph bar illustrates the dysregulated proteins involved in inflammation or angiogenesis according to the GO Biological Process database (**A**,**B** respectively). The volcano plot illustrates the differential expression of proteomics data. The dots represent different proteins according to the statistic *p*-value (y-axis) and relative abundance ratio (Log_2_ fold change) between ABCB5-positive and negative epithelial cells. The bold line shows the *p*-values cut-off and dysregulated proteins. upregulated in red and downregulated in green (**C**).

**Table 1 cells-12-01731-t001:** Key changes in vascular network of ABCB5KO mouse at different time points. ↑ Refers to significant increases in number of vessels.

Mouse Age	Changes in Vascular Network
4 weeks	↑Blood vessel sprouts ↑lymphatic vessel endpoints ↑lymphatic vessel branching points
8 weeks	↑Lymphatic vessel endpoints
26 weeks	No difference

## Data Availability

Publicly available proteomics datasets were analyzed in this study. This data can be found here https://www.ebi.ac.uk/pride/. Project Name: ABCB5+ limbal epithelial stem cells inhibit developmental but promote inflammatory (lymph)angiogenesis while preventing corneal inflammation. Project accession: PXD041004. Project DOI: Not applicable. Reviewer account details: Username: reviewer_pxd041004@ebi.ac.uk. Password: PADhlDRx.

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
