# Peer review of "ABCB5+ Limbal Epithelial Stem Cells Inhibit Developmental but Promote Inflammatory (Lymph) Angiogenesis While Preventing Corneal Inflammation"

_cells, 2023, doi:10.3390/cells12131731_

Round 1
Reviewer 1 Report
The authors have investigated the role of limbal epithelial stem cells in developmental as well as inflammatory angiogenesis / lymphangiogenesis and inflammation. Specifically, they have studied the consequences of ABCB5+ inactivation in mice, which is a marker protein expressed by such cells. Moreover, they have carried our proteomics analysis of human ABCB5+ vs. ABCB5- limbal epithelial cells. Based upon their results, the authors conclude that ABCB5+ limbal epithelial cells inhibit developmental angiogenesis, but on the other hand promote inflammatory angiogenesis / lymphangiogenesis while preventing corneal inflammation.
The study is well performed, the results are convincing and, in principle, support the conclusions drawn by the authors. Thus, the study provides useful information regarding the role of limbal epithelial cells in regulating developmental vs inflammatory angiogenesis / lymphangiogenesis in the limbus of the eye.
Criticism:
My major concern is that the authors have used a separate colony of C57BL/6 mice as control for their experiments. While it is certainly convenient to keep KO mice and WT control mice as separate colonies, there is a certain risk that the genetic background of KO and WT mice is not identical, which might lead to phenotypic differences that are not caused by the targeting of the specific gene of interest. Therefore, comparison of KO mice with heterozygous or wt littermates is the preferred approach. Did the authors perform such studies? What exactly is the source of the WT control strain? Were the KO mice and control mice back-crossed with the same C57Bl/6 substrain? Please provide detailed information.
Minor points of criticsm:
- Correct typo on page 2 line 80: assess
- Page 4 lane 165 should probably read as "magnetically", rather than "magically"
- Page 12 Figure 7 A,B: The words "Branche" and "Branchi" are not well chosen as titles.
- Correct typo on page 14 lane 504: Prostaglandin
- Correct sentence on page 17 line 644 "... in on ..."
Author Response
Dear Reviewer 1:
Please see the attachment.
Best regards

Reviewer 2 Report
The authors present a very interesting research on the topic of limbal stem cells. In the last 20 years many researchers investigated LESC, however there are many unanswerd questions, while the thechology already entered the clinical use. The authors investigated ABCB5 marker for LESC and performed complex analyses regarding the role of ABCB5 cells on inflammation and angyogenesis.
A few errors need to be corected:
Line 56 - Recently - lower case needed. Please move references 1,2 at the end of the phrase
60-62 - It is hard to recognize data supporting the hypothesis (ABCB5 LESCs role in angiogenesis). Better references needed.
Line 163 - please explain of change "magically separated"
Figure 1. C and D are also 4 weeks mice (E,F 8 weeks, G,H 26). 2D is I,J and 3D is K,L. Please verify throughout the text Figure 1 reference.
Line 347 - reference to figure 3 ? C and D are closeups, not inflammation corneas
Line 382- Figure 5F , maybe? Or better, change in Figure 5 the label (from F to E)
Lines 385-392 - many repeating statements, please revise
Line 396-405 - positive/negative ABCB+ a little confusing. Maybe change isotype with ABCB5+pos/neg
Line 516- aniridia is a rare disease (not a major cause of blindness). MAybe: LESC absence in congenital aniridia causes early childhood blindness.
Line 538 - E.18.5 and P10- please explain
References - 1st is missing (the numbering)
The english language is fine and apart from the small errors identified above the text is well written.
Author Response
Dear Reviewer 2,
Please see the attachment.
Best regards
Berbang Meshko

Round 2
Reviewer 1 Report
The authors have satisfactorily addressed the reviewer criticism.